# The Generation and Effects for Recyclable Waste from Households in a Megapolis: A Case Study in Shanghai

**Chaojie Yu [1], Junqing Xu [1], Aihua Zhao [2], Peiyuan Xiao [1], Jun Tai [3], Zhujie Bi [3] and Guangming Li [1,*]**

1   College of Environmental Science and Engineering, Tongji University, Shanghai 200092, China; 2030524@tongji.edu.cn (C.Y.); xjq0512@tongji.edu.cn (J.X.); zsdxpy@163.com (P.X.)
2   Shanghai Chengtou Group Corporation, Shanghai 200020, China; 13701829031@163.com
3   Shanghai Environmental Sanitary Engineering Design Institute Co., Ltd., Shanghai 200232, China; taij@huanke.com.cn (J.T.); bizj@huanke.com.cn (Z.B.)
*   Correspondence: ligm@tongji.edu.cn

**Abstract:** Shanghai is one of the world-leading megapolises facing the challenge of ecological sustainable development. The recyclable waste from households (RWH) generated in Shanghai has increased rapidly since the implementation of garbage classification in 2019. However, there are no rigorous data on the generation and collection of RWH, and the corresponding countermeasures are required to be studied. This paper attempted to investigate RWH generation and identify the effects of RWH recycling in Shanghai. We used questionnaires combined with a field survey to investigate the competent authorities and leading recycling enterprises to analyze the characteristics of RWH generation. We conducted a monthly survey of 52 leading recycling enterprises in 11 typical districts for 2020. We also identified the main influencing factors of RWH generation using a multiple linear regression model. In addition, we popularized the model to estimate Shanghai's RWH generation rate. Results show that data from leading recycling enterprises surveys were more accurate and reached a maximum of 82,104.77 kg/cap/month in November 2020. Higher RWH generation was found in suburban districts at 36,396.20 kg/cap/month. Shanghai's RWH generation rate was 6253.60 t/d through model calculation. The educational level of household managers, regional economic condition, resident population, and disposable income impact RWH generation. Based on the abovementioned results, the implications for RWH management were discussed. We propose to promote the combination of theoretical simulation and information data platform construction. Meanwhile, it is also necessary to improve the capacity of the collection and transport system and accelerate the construction of pre-treatment bases in Shanghai.

**Keywords:** ecology; recyclable waste from households; generation; recycling; effect; Shanghai

## 1. Introduction

In 2015, the United Nations introduced the Sustainable Development Goals [1]. One of this concept's critical issues is ensuring a stable ecological environment and sustainable consumption. Ecological behavior is closely related to resource structure, the energy industry, and consumption patterns [2–4]. Meanwhile, many studies recognize the need to measure the environmental impact of ecological behavior [5], and promoting recycling is considered an important way for household waste disposal to realize sustainable ecological development [6,7].

In recent decades, as the economy grew rapidly and living conditions improved, increasing amounts of household garbage were produced [8,9]. The disorganized management and ineffectiveness of household garbage disposal [10–12] have resulted in serious environmental contamination and resource depletion. Open dumping of untreated household waste causes gas emissions and chemical contamination in water and soil [13]. Traditional landfill treatment of household waste accounts for a smaller market share. People reuse

resources to achieve ecologically sustainable development, including mechanical regeneration and energy recovery, such as pyrolysis, gasification, and biomethanation [14,15]. Before resource utilization, the critical link is to recycle the recyclable components of household waste. In China, recycling is currently seen as a crucial means of addressing the issue of excessive household garbage and safeguarding human health and the environment [16–18]. Consequently, understanding the features and impacts of RWH generation is essential for the recycling and management of RWH [19,20].

RWH has been considered and studied by many countries. For example, Ref. [21] conducted a life cycle assessment to identify the strengths and weaknesses of the Swedish RWH system. The RWH system was found to gain more environmental benefits through active and efficient sorting and recycling measures. Ref. [22] analyzed 1292 samples from 120 households in Mexicali, Mexico, and found that the RWH generation rate in an urban community is higher than in rural communities (0.470 kg/day and 0.310 kg/day, respectively). The recycling program would reduce waste generation and significantly extend the lifetime of recyclables.

Comparatively, the research on RWH in China started in recent years. Although China implemented the policy of the separation of recyclables earlier, the recycling system has not kept pace [23], resulting in the recycling of RWH in China lagging behind that of developed countries [24,25]. In January 2022, the National Development and Reform Commission issued the "Guidance on Accelerating the Construction of the Recycling System of Waste Materials" [26]. The recyclables recycling and resource utilization industry has been gradually regulated. However, due to the lack of accurate generation data, the quantity and quality of recyclables are unclear [27,28], and the research on RWH generation in China is still preliminary [17,29,30]. It is vital to improve RWH recycling development by determining the number of RWH generations and establishing databases.

According to the general components of municipal waste, RWH can be divided into glass, metal, plastic, paper, fabric, and other categories [31]. The mechanism of the driving forces influencing RWH generation tends to be regionally different. Managing the massive RWH has inevitably become a tough challenge for the Chinese government [32,33]. It is suggested that local governments need to make appropriate plans according to local conditions and economic and social development to determine the mechanism.

Therefore, we performed a case study in Shanghai, the first city in China to implement mandatory household waste classification [34], to quantify the RWH generation characteristics. The government cooperates with leading recycling enterprises to obtain recycling data through intelligent facilities and gradually standardizes the management of RWH. Shanghai is expected to take the lead in realizing the precise management of RWH. A systematic survey of 52 leading recycling enterprises in 11 typical districts was conducted to investigate the amount of RWH recycled monthly in 2020. The RWH generation rate was estimated by establishing a multiple linear regression model. Moreover, we propose strategies for RWH management in Shanghai, which can also serve as references for other cities in China to promote RWH management in the future.

This study was divided into four sections. The background of this study was introduced in the "Introduction" section. In the "Materials and Methods" section, we introduce the case area and investigation methods in Section 2.1, parameter selection and establishment process of recovery mathematical model in Section 2.2, and data analysis and processing methods in Section 2.3. In the "Results and Discussion" section, we conduct characterizations of recyclable waste from households in Section 3.1; the driving factors affecting RWH generation in the mathematical model in Section 3.2; the popularization of the model in Section 3.3, and the RWH management suggestions in Section 3.4. Finally, we summarize the research content and significance of this research and put forward the limitations of the research and the focus of future work in the "Conclusions" section.

## 2. Materials and Methods

Field surveys and questionnaires are practical approaches to information collection on waste generation [35]. This study obtained first-hand data through field surveys and questionnaires with leading recycling enterprises in Shanghai. We formed a preliminary understanding of RWH generation and its effects since implementing compulsory garbage classification in Shanghai.

### 2.1. Case Area and Investigation Methods

#### 2.1.1. Study Area

Shanghai is a megapolis located on the east coast of China, covering an area of 6340 km$^2$ (Figure 1). Shanghai is an essential economic engine of China, whose GDP reached 3815.532 billion yuan in 2019 [36]. There are 24.28 million permanent residents, with an average population of 2.64 people per household [36].

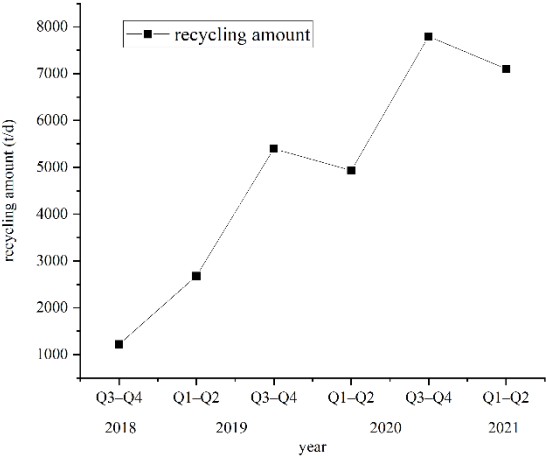

**Figure 1.** Daily average variation of RWH recycling volume in Shanghai.

Recently, sustainable development has become the first development goal of international megalopolises. The pursuit of sustainable development was defined in the Brundtland Report as "meeting the needs of contemporary humanity without losing the ability of future generations to meet their own needs" [37] in 1987. John Elkington proposed the "triple bottom line" [38] of sustainable development: environmental, economic, and social. According to [39], household waste has gradually become a problem that needs to be addressed in the urban development of Shanghai in recent years. The compulsory classification provision of Shanghai residents' domestic waste since 2019 has been successful, and local policymakers emphasized educating the public about "how to" instead of "why to sort wastes". With effective waste classification, the net carbon emission of separated processing was reduced to 0.11 ton CE/ton waste [35].

Our investigation of the government and enterprises found that Shanghai's compulsory waste classification work has achieved remarkable outcomes in the past two years, with 95% of residents meeting the waste separation standards [40] by the end of 2021. A domestic waste classification system has been built depending on the information platform. However, the recycling of RWH is still in the primary stage. There are still problems, such as that the recycling classification is not fine enough and there is a low recycling utilization rate.

Many cities in China have experienced uneven urban expansion, forming a pattern of co-development of urban and suburban areas, and Shanghai has followed this pattern. Therefore, we selected five typical urban districts and six typical suburban districts to analyze the difference in RWH generation and category between urban and suburban areas.

### 2.1.2. Investigation Method

Surveys about RWH authorities and RWH leading recycling enterprises were carried out. We investigated Landscaping & City Appearance Administrative Bureaus of 11 districts in Shanghai and 52 RWH leading recycling enterprises in these 11 districts through online questionnaires and face-to-face surveys. We surveyed five urban districts and six suburban districts to balance numbers between different regions.

Each district's Landscaping & City Appearance Administrative Bureau is in charge of RWH management and the construction of the RWH collection and transportation system. We obtained the official RWH generation data from these departments and the construction of a "point-station-field" RWH recycling system in Shanghai (Table 1).

**Table 1.** Recycling sites' distribution in different districts in Shanghai, 2020.

| District | Area of Region (km$^2$) | Permanent Residents (10,000) | Number of Recycling Sites | Site Area Coverage (per/km$^2$) | Site Population Coverage (per 10,000 People) |
|---|---|---|---|---|---|
| 1. Urban | | | | | |
| Hongkou district | 23.45 | 75.75 | 1218 | 33.09 | 10.24 |
| Jing'an district | 37.37 | 97.57 | 1183 | 27.40 | 10.49 |
| Changning district | 38.3 | 69.31 | 836 | 24.54 | 13.56 |
| Yangpu district | 60.61 | 124.25 | 659 | 19.30 | 9.42 |
| Putuo district | 55.53 | 123.98 | 452 | 15.65 | 7.01 |
| 2. Suburban | | | | | |
| Baoshan district | 270.99 | 223.52 | 776 | 4.49 | 5.45 |
| Minhang district | 370.75 | 265.35 | 1024 | 3.19 | 4.46 |
| Songjiang district | 605.64 | 190.97 | 940 | 1.38 | 4.38 |
| Fengxian district | 687.39 | 114.09 | 1170 | 0.96 | 5.78 |
| Jinshan district | 586.05 | 82.28 | 869 | 0.77 | 5.49 |
| Chongming district | 1413.00 | 63.79 | 776 | 0.27 | 5.99 |

Data source: The area of region and number of permanent residents are from http://tjj.sh.gov.cn/7rp-pcyw/20210519/f470438f902f4c88af63be0f84c9808f.html (accessed on 18 May 2021).

The leading recycling enterprises in each district are directly responsible for RWH recycling. In 2018, Shanghai identified several recycling firms through bidding and selection in compliance with applicable legislation. It established a "two-network" market structure (integration of urban sanitation system and recycling system) with 64 leading recycling enterprises. The RWH generation data reported by these enterprises through the information platform are the closest to the actual situation. We obtained the monthly recycling data of different categories of RWH (glass, metal, plastic, paper, fabric) in 2020 from these leading recycling enterprises.

### 2.1.3. Questionnaire Design

Two questionnaires were designed. For Questionnaire A, "Survey on RWH generation and Recycling System in Shanghai", the target groups were Landscaping & City Appearance Administrative Bureaus in each district of Shanghai. For Questionnaire B, "Survey on Recycling Amount of Leading Recycling Enterprises in Shanghai", the target groups were the leading recycling enterprises in each district of Shanghai. After we prepared the draft, we underwent an intensive discussion with relevant municipal government leaders and experts. The investigation team then conducted a pilot survey to test the accuracy of the questionnaires.

Questionnaire A aimed to investigate the official data of RWH generation and identify the main reasons for its inaccuracy. The construction of the RWH collection and transportation system also needed to be studied. The main questions in Questionnaire A included: 1. the official RWH recycling amount (since the implementation of the regulations); 2. the main categories of RWH; 3. the leading enterprises' information of each district; 4. the number of recycling sites in each district. Questionnaire B aimed to investigate the overall recycling amount of RWH and each subdivided category. The main questions in Questionnaire B included: 1. the RWH recycling amount of enterprises in each month of 2020; 2. the

proportion of RWH categories; 3. the main direction of resource utilization of RWH in each category (downstream enterprises).

Sixteen copies of Questionnaire A were issued for the research, and 11 were received. Questionnaire B was issued to the leading recycling enterprises in these 11 districts, with 52 copies, all of which received replies.

### 2.2. Mathematical Modeling

A multiple linear regression model aims to study the influence of two or more independent variables on a dependent variable under linear correlation conditions. However, many factors affect RWH generation. Regression analysis could accurately measure each influencing factor to optimize the prediction, so the multiple regression analysis is a more convenient and accurate path to identify the driving forces of RWH generation.

Generally, municipal solid waste generation is mainly affected by population, economic development, residents' living standards, infrastructure construction, and other factors. For RWH generation, this study chose per capita disposable income, GDP per unit area, college education population, and permanent residents as driving factors. According to the preliminary investigation, Changning district has a different recycling mode ("supply and marketing cooperative" mode) from other districts. Moreover, the data on leading recycling enterprises in Yangpu district are incomplete. Thus, the remaining nine typical districts (Table 2) were selected as the analysis objects in the model analysis to obtain accurate results.

**Table 2.** Driving factors in different districts in Shanghai, 2020.

| District | Per Capita Disposable Income | GDP Per Unit Area | College Education Population | Permanent Residents (10,000) |
|---|---|---|---|---|
| Hongkou district | 83,256.00 | 44.66 | 0.41 | 75.75 |
| Jing'an district | 85,625.00 | 62.79 | 0.41 | 97.57 |
| Putuo district | 81,386.00 | 20.34 | 0.41 | 123.98 |
| Baoshan district | 71,456.00 | 5.37 | 0.32 | 223.52 |
| Minhang district | 74,764.00 | 6.88 | 0.38 | 265.35 |
| Songjiang district | 59,515.00 | 2.70 | 0.16 | 190.97 |
| Fengxian district | 49,439.00 | 1.65 | 0.21 | 114.09 |
| Jinshan district | 48,010.00 | 1.76 | 0.20 | 82.28 |
| Chongming district | 41,990.00 | 0.27 | 0.13 | 63.79 |

Data source: http://tjj.sh.gov.cn/7rp-pcyw/20210519/f470438f902f4c88af63be0f84c9808f.html (accessed on 18 May 2021).

The Pearson's product-moment correlation coefficient [41] in this model can be expressed as follows in Equation (1):

$$\beta_m = \frac{\sum_{i=1}^{n} X_{mi} y_i - \left(\sum_{i=1}^{n} X_{mi}\right) \times \left(\sum_{i=1}^{n} y_i\right)}{\sqrt{\left[n \times \sum_{i=1}^{n} X_{mi}{}^2 - \left(\sum_{i=1}^{n} X_{mi}\right)^2\right] \times \left[n \times \sum_{i=1}^{n} y_i{}^2 - \left(\sum_{i=1}^{n} y_i\right)^2\right]}} \tag{1}$$

where $\beta_m$ represents the correlation coefficient of $y$ concerning $X_m$; $X_{mi}$ represents the value of $X_m$ corresponding to the ith data; $y_i$ represents the value of recycling corresponding to the ith data, in tons; m is the value of selected influence factors, and is set to 4 in this study; $n$ is the value of valid data, and this study selected nine typical districts as data sources.

Per capita disposable income, GDP per unit area, college education population, and permanent residents were preliminarily taken as independent variables, and the RWH recycling capacity of leading recycling enterprises was taken as a dependent variable in this model, which can further be expanded as follows in Equation (2):

$$Y = P_0 + \sum_{i=1}^{4} P_i \times X_i \tag{2}$$

where Y represents the RWH recycling capacity of leading recycling enterprises; $P_0$ is an intercept; $P_i$ represents the regression coefficient.

Based on driving factor data of 9 districts and the RWH recycling volume of leading recycling enterprises in 2020, the influence factor coefficients of the multiple linear regression equation could be calculated by substituting the above formula.

In order to predict the overall RWH generation in Shanghai, we extrapolated the mathematical model by introducing correction factors. Specific discussions can be found in Section 3.3.

### 2.3. Statistical Analysis

SPSS 25.0 [42] was used to perform a correlation analysis and investigate the correlation between the RWH generation rate and the test items. Regression and multiple regression analyses were then conducted to create a multivariable linear equation describing the RWH recycling rate and the test items. The equations can serve as a reference for testing RWH generation in Shanghai and other cities.

## 3. Results and Discussion

### 3.1. Characterization of Recyclable Waste from Households

#### 3.1.1. Generation Rate

Since implementing "The Shanghai Municipal Household Garbage Management Regulations", the construction of the whole process system of recyclables in Shanghai has been promoted in an orderly manner. The daily volume of RWH generation has shown an overall upward trend. According to our research data from Shanghai Landscaping & City Appearance Administrative Bureaus, in the first half of 2021, the average daily RWH collection volume in Shanghai was approximately 7104 tons per day, an increase of 65% compared with the same period in 2019 (Figure 1). The data for RWH recycling in the first half of 2021 and the first half of 2020 show a slight decrease compared to the previous period. The main reason is that the total amount of household garbage during the corresponding period decreased due to the impact of COVID-19 [43].

The field research of relevant government departments found that Shanghai's official RWH generation data are provided by the city's leading recycling enterprises and other market-based enterprises. The city's leading recycling enterprises declared eighty percent of the data, and market-based enterprises in each district summarized 20 percent. In the actual statistical process, the average daily RWH generation data may be greater than the actual situation due to the difficulty of approving the fragmented sources of recycling volume and the lack of adequate supervision of the information uploaded by each transfer station.

The accuracy of official data in previous years was not high due to the unclear recycling subjects and lack of adequate supervision. With the further promotion of the standardized management of leading recycling enterprises in Shanghai and the improvement of the construction level of the information platform, the RWH recycling data of leading recycling enterprises have higher accuracy. Therefore, a more accurate understanding of the actual RWH generation in Shanghai can be obtained by directly investigating the RWH data of 52 leading recycling enterprises in 11 districts.

We analyzed the monthly changes in RWH in 2020 and the differences between urban and suburban areas (Figure 2a,b). The largest quantity of RWH was generated in November (82,104.77 kg/cap/month). The possible explanation is that discounts and promotions attract residents during the "Double 11" shopping festival. People's willingness to consume has been dramatically improved, and the production of express packaging such as paper boxes, plastic bags, and shockproof foam is increasing RWH generation [44]. Overall, with the gradual standardization of recycling and the improvement of residents' awareness, RWH generation has shown a rapid rise since the outbreak of COVID-19 in February 2020 and has been at a very high level in the last quarter of 2020 (75,987.82 kg/cap/month in October and 75,824.30 kg/cap/month in December, relatively).

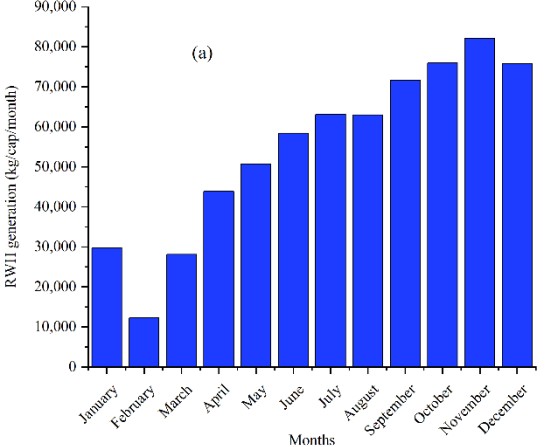
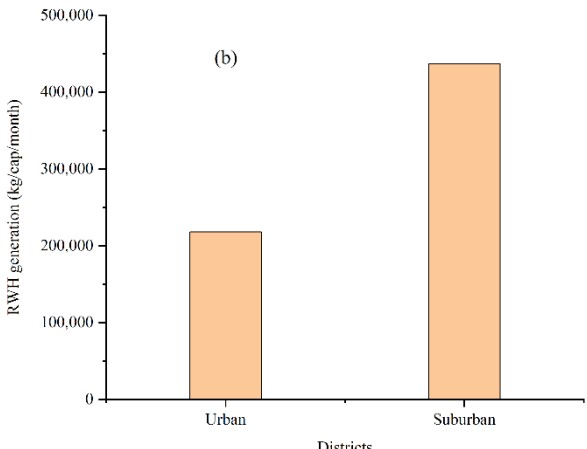

**Figure 2.** (**a**) RWH generation on each month in 2020; (**b**) RWH generation in urban and suburban districts.

Higher RWH generation was found in suburban districts (18,164.70 kg/cap/month in the urban districts and 36,396.20 kg/cap/month in the suburban districts) (Figure 2b), which is consistent with existing results that more RWH is generated with a large area and more permanent residents.

### 3.1.2. Physical Categories

According to the RWH quantity and quality investigation in each district of Shanghai, RWH is mainly divided into glass, metal, plastic, paper, fabric, and other categories. Figure 3a,b shows the proportion of recyclables of different categories in urban and suburban areas of Shanghai. Paper, metal, and plastic products are widely used in our daily lives, so, whether in suburban or urban areas, these three types of RWH accounted for a relatively high proportion of all categories, especially paper products.

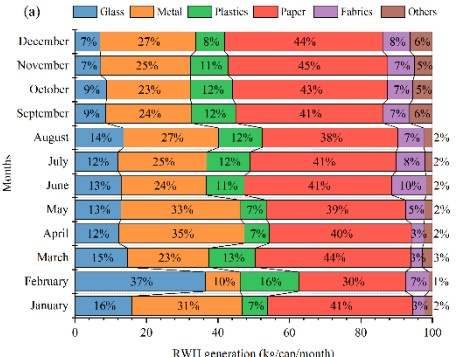
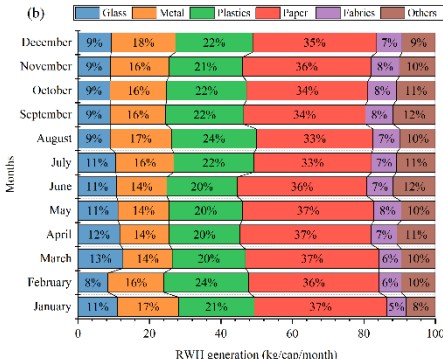

**Figure 3.** (**a**) The proportion of different RWH categories of urban districts in 2020; (**b**) the proportion of different RWH categories of suburban districts in 2020.

It can be found that within the statistical scope, the structure of recyclable articles in urban areas is unbalanced, with waste paper and waste metal as the central part, and the proportion of other types is less than 10%. The difference in the RWH ratio between suburban and urban areas is mainly due to the different lifestyles and consumption habits of residents. For example, in urban areas, where white-collar workers are concentrated, most of the waste generated by offices is paper products. Therefore, the proportion of waste paper in urban areas is higher than that in suburbs. In addition, due to lax regulations, plastic products (including single-use plastics) are more widely used in suburban areas, so the proportion of waste plastics in RWH is higher in suburban areas than in urban areas. Compared with the structural proportion of RWH in urban areas, the proportion of RWH in suburban areas is more balanced—mainly waste paper, waste metal, and waste plastic,

being much lower than the proportion of waste paper, waste metal, and waste plastic in urban areas. By comparing the various RWH categories in different areas, it can be seen that the variation range of RWH in suburban districts is lower than that in urban Shanghai and more stable.

According to the investigation of the downstream locations of recyclables of leading recycling enterprises, we found that the utilization enterprises are mainly distributed in the Yangtze River Delta region.

### 3.2. Factors Influencing RWH Generation

The driving factors of the RWH generation rate are explored in this section. An ANOVA and rank correlation analysis were applied in this study. It was found that significant correlations existed between the RWH generation rate and the factors of per capita disposable income, the GDP per unit area, college education population, and permanent residents (Table 3). Moreover, no multicollinearities were found among the factors mentioned above.

**Table 3.** Linear regression model of RWH generation rate.

| Variable | Coefficients (Unstandardized) | | t-Stat | Sig | $R^2$ | Durbin–Watson |
|---|---|---|---|---|---|---|
| | Coefficients | Standard Error | | | | |
| Constant | 179,949.366 | 44,141.729 | 4.077 | 0.027 | | |
| Per capita disposable income | −7.245 | 3.032 | −2.389 | 0.097 | | |
| GDP per unit area | 2458.341 | 723.796 | 3.396 | 0.043 | 0.918 | 2.007 |
| College-educated population | 625,235.114 | 151,094.127 | 4.138 | 0.026 | | |
| Permanent residents | 1002.043 | 379.660 | 2.639 | 0.078 | | |

The linear regression equation of RWH recycling of leading recycling enterprises could be concluded as:

$$Y = -7.245 \times X_1 + 2458.341 \times X_2 + 625235.1 \times X_3 + 1002.043 \times X_4 + 179949.366 \quad (3)$$

(1) Per capita disposable income

It can be seen from the above regression results that per capita disposable income has a negative effect on the RWH recycling of leading recycling enterprises. In other words, every 1% increase in per capita disposable income will bring a 7.245% inhibition to the RWH recycling of leading recycling enterprises. Due to the increase in income level, people are more inclined to use better-quality products instead of disposable products. At the same time, the improvement of environmental awareness will also prolong the service life of products, making them more challenging to be discarded and converted into RWH. Ref. [45], concluding that, with the increase in personal disposable income, the per capita amount of RWH treatment decreased at first, then increased in the second stage, and finally decreased, presenting an inverted N-shaped curve. This law was more significantly reflected in urban areas than in rural areas.

(2) GDP per unit area

According to the regression results, GDP per unit area has significant positive benefits for the RWH recycling of leading recycling enterprises. RWH generation in cities is significantly associated with economic and social development [46]. The more developed a city is, the higher the waste pressure caused by natural forces. Two overlapping reasons cause this phenomenon. On the one hand, in places where the pace of work is fast, people's consumption level is high, and they rely heavily on fast food delivery. The shift in societal growth will result in a significant increase in RWH. On the other hand, residents of developed cities are less likely to participate in the recycling system of scavengers [47], and some low-value recyclables are discarded, leading to an increase in RWH production.

(3)  College-educated population

According to the regression results, the college-educated population also has significant positive benefits for the RWH recycling of leading recycling enterprises. Indirectly, the amount of paper in home garbage might indicate a city's economic and educational development [48]. The more economically and educationally developed a city is, the higher the proportion of high-value materials in its municipal household waste. The discrepancy between the percentage of paper in MSW in high-income nations and low-income countries is even more severe, with the share of paper in MSW in high-income countries ranging from 25 to 40 percent. It is only 2 to 6 percent in low-income countries [49].

(4)  Permanent residents

Permanent residents have significant positive benefits for the RWH recycling of leading recycling enterprises. For megapolises such as Beijing and Shanghai, there are many permanent residents, and the amount of RWH generated every day is also striking. Accelerating the construction of permanent recycling centers, innovating the recycling mode, and improving the statistical level of recycling information are all optimal paths to improve RWH recycling [50].

*3.3. Model Popularization for RWH Generation*

According to the ANOVA results of the prediction model, F = 18.889, *p* = 0.018 < 0.02 < 0.05, it can be considered that the regression model has passed the F-test and the fitted equation is statistically significant. A model can make reasonable predictions if the $R^2$ is more than 0.35 [51]. The value of $R^2$ (0.918) (Table 2) demonstrated that the model developed here was within the range of fitness. Generally speaking, the Durbin–Watson test values are distributed between 0 and 4, and the closer they are to 2, the more likely they are to be independent of each other [52]. The Durbin–Watson test value of this prediction model is 2.007, so it can be considered that the observed values in this study have good mutual independence.

The multiple regression prediction models of the RWH recycling of leading enterprises mainly explain the recycling volume of RWH in Shanghai, excluding the influence of social recycling methods. In order to further predict the overall generation volume of RWH in Shanghai, the model needs to be modified in combination with the ratio of recyclables' recycling amount to total recyclables' production and the collection volume of leading enterprises.

The revised formula is:

$$Y_{\text{all}} = P_0 + \sum_{i=1}^{4} P_i \times X_i \times \varphi \tag{4}$$

$P_i$ is the regression coefficient, and $\varphi$ is the correction factor.

The data of leading enterprises in Minhang district obtained in this study are relatively perfect; combined with the specific data of the whole means of recyclables in Minhang district, the typical value of the correction factor $\varphi$ can be determined by the following formula:

$$\varphi = \frac{Y_0}{Y_{\text{leading enterprises in Minhang}}} \tag{5}$$

In the formula, $Y_{\text{leading enterprises in Minhang}}$ represents the amount of RWH collected by the representative subject enterprises in Minhang district, and $Y_0$ represents the total amount of RWH collected in this district. After calculation, the value of $\varphi$ amounts to 1.489. Then, the linear regression equation of RWH recycling of Shanghai can be obtained as:

$$Y = -10.788 \times X_1 + 3660.470 \times X_2 + 930975.064 \times X_3 + 1492.042 \times X_4 + 223274.606 \tag{6}$$

The revised multiple linear regression prediction models of Shanghai can be calculated using the data of relevant impact factors in 2020. The result of the revised model is that

6253.60 t/cap/day of RWH will be generated in 16 districts of Shanghai (Table 4). Compared with the official data (6375 t t/cap/day) of RWH generation in 2020, the predicted value is 98.10% of the actual recycling amount, which has certain reliability. This prediction model can use the impact factor data of the research year to predict the number of recyclables produced in Shanghai and each administrative region.

**Table 4.** Predictions form different multiple linear regression prediction models for RWH generation.

| Data Source | Multiple Linear Regression Prediction Model for RWH Generation of Leading Enterprises | Multiple Linear Regression Prediction Model for RWH Generation of Shanghai |
|---|---|---|
| Area | Forecasted daily average value of leading enterprises (t/d) | Forecasted daily average value of Shanghai (t/d) |
| Hongkou district | 39.69 | 59.10 |
| Qingpu district | 169.72 | 252.72 |
| Minhang district | 430.97 | 641.71 |
| Baoshan district | 274.92 | 409.35 |
| Jiading district | 299.94 | 446.61 |
| Jinshan district | 113.29 | 168.69 |
| Songjiang district | 122.89 | 182.99 |
| Fengxian district | 194.16 | 289.11 |
| Chongming district | 51.92 | 77.31 |
| Huangpu district | 668.43 | 995.29 |
| Xuhui district | 194.92 | 290.24 |
| Changning district | 41.78 | 62.21 |
| Jing'an district | 166.40 | 247.76 |
| Putuo district | 43.11 | 64.19 |
| Yangpu district | 130.84 | 194.82 |
| Pudong district | 1256.88 | 1871.50 |
| Total | 4199.87 | 6253.60 |

*3.4. Management of Recyclable Waste from Households*

The results were extrapolated to assess RWH generation and future collection and management in Shanghai.

Recycling RWH is challenging, and there is an urgent need to implement policy support. Recycling RWH presents the most significant difficulty in municipal waste collection. Paper, metals, and other recyclables with high value-added are recycled in a scattered manner. However, the recycling rate for plastics and other recyclables with low value-added is still meager [53]. It is difficult for RWH recycling businesses to generate a profit. The number of subsidies and other policies must be controlled; nevertheless, there are no systematic and distinct support programs. Due to the high expense of recycling and limited profitability of reuse, a portion of waste plastics, waste glass, waste wood, and other low-value trash is frequently combined with dry garbage [54]. Low earnings are combined with dry waste incineration, which hinders the exploitation of resources on the back end.

The coverage of RWH recycling businesses is insufficient, and the markets for RWH are unstable [55]. First, the types and quantities of RWH recycling businesses are inadequate and insufficient. Most of Shanghai's businesses specialize in industrial solid trash, building garbage, and electronic waste. RWH such as waste cloth and miscellaneous plastic are difficult for businesses to landfill [29]. Second, some RWH are transported to neighboring provinces for disposal under Shanghai's stringent environmental standards. With the expanding environmental protection standards in other provinces, however, the disposal of these pollutants might pose significant difficulties.

The capacity for proper sorting of RWH is limited, and the closed-loop recycling system has not yet been constructed. Currently, Shanghai's fine sorting capability is restricted. The primary reasons for this are as follows. Firstly, the development of sorting sites must be expedited. Shanghai has built a new sorting system in Changning district, Yangpu district, Minhang district, Baoshan district, Pudong district, Songjiang district, Jiading district, Fengxian district, and Chongming district at this time. However, huge

sorting sites with detailed sorting features are still lacking. Secondly, the capacity of sorting businesses must be enhanced. The business of sorting has reached a fever pitch. Recycling has flourished due to waste sorting, and several Internet firms have entered the market [56]. Several clever sorting and recycling equipment manufacturers and operators have entered the recycling business. Several businesses that make and run sophisticated sorting and recycling equipment have arisen. However, many companies have only been in business for a short time and have a low market share. Due to their inexperience in the industry, lack of technical innovation, and inability to sort waste in the middle-end, these businesses are limited to sorting and recycling in the community, business buildings, and other locations [30]. Furthermore, they cannot package waste for resale or sell it in large quantities.

Based on our investigation, Shanghai has piloted the construction of whole-process garbage classification information platforms in Songjiang district and Jiading district, and the "Internet + Recycling" model is gradually maturing. We suggest that the combination of theoretical simulation and information platform construction [57] should be continuously promoted to improve the ability to accurately grasp the quantity, quality, and transportation situation of recyclables, ensure the effectiveness of trend analysis and strategy research, and achieve accurate statistics and efficient prediction of recyclables collection and transportation data. In the future, the theoretical model and information platform can be used to realize data tracking and traceability of recyclables, provide big data support for the layout and construction of the resource-based industry, and significantly improve the efficiency of management and resource-based processing.

Due to the restriction of the land use index, regional environmental impact assessment, and other factors in Shanghai, it is difficult for recycling enterprises to be established, and the recyclables in Shanghai are still mainly transported to other places for disposal. We suggest constructing a recycling pretreatment based in Shanghai. It will be helpful to identify the different types of recyclable waste from households, as well as the high-value potential applications of the material preparation of semi-finished raw materials in the downstream industry [58]. This is in line with Shanghai's facility construction concept and also can promote the formation of a collaborative resource-based treatment system for recyclables within the Yangtze River Delta.

## 4. Conclusions

A comprehensive survey on recyclable waste from household generation and collection in Shanghai was conducted. Questionnaires from RWH authorities and leading recycling enterprises were analyzed, and a preliminary understanding of the quality and quantity of RWH and the recycling system in Shanghai was formed.

A mathematical model of RWH generation ($R^2$ = 0.918) was developed, which indicated that the per capita disposable income, GDP per unit area, college-educated population, and permanent residents influence the RWH generation rate.

This research can provide scientific support for improving government decision-making, e.g., breaking through the information barriers of the government and enterprises, focusing on the integration of model simulation with the data platform, expanding the capacity of the collection system, and setting up pretreatment bases in Shanghai. The experience and suggestions in this study can also serve as a reference for other Chinese cities to improve the efficacy of RWH recycling.

We are aware that our research might have limitations regarding the survey scope and the research time. We have not completely collected the recycling data of leading enterprises in 16 districts of Shanghai. Moreover, although the time frame of our study covers monthly changes in 2020, the effect of chance cannot be excluded. Future work needs to focus on data collection and a longitudinal study. With further data collection and the extension of the research time, the recycling law and the impact on the ecological environment of RWH can be better clarified.

**Author Contributions:** Conceptualization, C.Y. and G.L.; Methodology, C.Y., J.X. and P.X.; Software, J.X. and P.X.; Investigation, Z.B. and J.T.; Data curation, Z.B., J.T. and A.Z.; Writing—original draft preparation, C.Y.; Writing—review and editing, C.Y., J.X., P.X., A.Z. and Z.B.; Supervision, G.L., J.T. and A.Z. All authors have read and agreed to the published version of the manuscript.

**Funding:** This research was funded by the Ministry of Science and Technology of the People's Republic of China (2018YFC1900700).

**Institutional Review Board Statement:** Not applicable.

**Informed Consent Statement:** Informed consent was obtained from all subjects involved in the study.

**Data Availability Statement:** Not applicable.

**Acknowledgments:** We thank the Special Committee of Shanghai Municipal People's Congress and the leaders of relevant governments, enterprises and university experts for their support and guidance for the research team of "Research on the Linkage Mechanism of Recycling and Utilization of Recyclable Waste from Households in Yangtze River Delta Region".

**Conflicts of Interest:** The authors declare no conflict of interest.

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
