# Peer review of "The Generation and Effects for Recyclable Waste from Households in a Megapolis: A Case Study in Shanghai"

_sustainability, doi:10.3390/su14137854_

Round 1

Reviewer 1 Report

Waste recycling is directly related to ecology, which is very important for taking care of the future of our planet. The manuscript submitted for review addresses an interesting and timely topic related to generating and reprocessing recyclable waste. The authors analyzed households and recycling companies in a selected area of China from this angle.  

The manuscript structure is correct. However, I have some comments for the authors.

The "Introduction" section is far too poor and does not provide enough theoretical background for the later sections of the manuscript. Segregation of waste at home and its subsequent recycling are directly related to ecology. Therefore, I propose to start this section with a brief general characterization of different environmental behaviors and their impact on the environment. You can use the following literature: https://doi.org/10.3390/en15051690 ; https://doi.org/10.1177/0013916517701796 ; https://doi.org/10.3390/en14061767  

Next, you can go into more detail about waste segregation and disposal options.

Consider adding "ecology"; "recycling" as keywords;

At the end of the "Introduction" section, please present the further structure of the manuscript with the characteristics of each section.

At the end of the "Conclusions" section, please present the limitations of the manuscript and opportunities for future research in this area.

Reviewer 2 Report

The topic discussed in the manuscript is an emergent one in terms of the urbanization and development in cities in China and similar countries around the world. The findings of the research will be useful in multiple ways, particularly for the management of the wastes generated in urban backdrop. The variables considered for highlighting the production and variations of the recyclable waste could have been more specific, still, the overall work is worthy for publication.  In my opinion, the research work will prove useful in monitoring and managment of the recyclable wastes generated in the urban context. I would just ask to provide a diversity assessment of the wastes generated in different months (figure 3). Minor revisions required in the article are marked in the attached pdf file. 

I find the overall merit of the article to be high.

Reviewer 3 Report

This paper makes a comprehensive survey on the recyclable domestic waste produced and collected in Shanghai, analyzes the questionnaires of the competent departments of recyclable domestic waste and the main recycling enterprises, and preliminatively understands the quality and quantity of recyclable domestic waste and the recycling system of Shanghai.  A monthly survey of 52 leading recyclers in 11 typical regions was conducted in 2020.  Multiple linear regression model was used to determine the main factors influencing the generation of recyclable household garbage.  In addition, the model is extended to estimate the power generation rate of recyclable domestic waste in Shanghai.  A mathematical model (R2=0.918) was established to show that per capita disposable income, GDP per unit area, college-educated population and permanent resident population have an impact on the generation rate of recyclable household garbage.  On this basis, the generalized model of Shanghai recyclable MSW production evaluation is established.  The analysis in this work will provide data to support government decision-making, for example, to promote the integration of theoretical simulation and information data platform construction.  At the same time, it is necessary to improve the capacity of collection and transportation systems and speed up the construction of a pretreatment base in Shanghai.  

However, this study still has the following shortcomings:  

1: Results and discussion section, the predicted value of multiple linear regression model is 98.10% of the actual value compared with the official RWH power generation data (6375 T /cap/day) in 2020.  The predicted value is only compared with the data of 2020, which lacks credibility. It is suggested to enrich the data and compare with the data of 2017,2018, and 2019.  

2. The questionnaire only published the results without mentioning the setting of the topic. It is suggested to publish the setting of the relevant topic to make the article more convincing.  

3. In the results and discussion section, the proportion analysis of recyclables of different categories in urban and suburban areas of Shanghai is not comprehensive enough. The proportion analysis of recyclables of different categories and the differences between urban and suburban areas are only described on the surface, but the reasons for this situation are not explained  

Therefore, this study can be published in this journal after modification. 
